# Benchmarking Transfer Learning: From Simple Baselines to Combined Scorers for Transferability Estimation

## Abstract

In the evolving landscape of deep learning, selecting the best pre-trained models from a growing number of choices is a challenge. Transferability scorers propose an efficient alternative to this challenge by calculating a proxy to rank a pool of pre-trained model candidates. Despite their promise, the field currently lacks standardized evaluation protocols, consistent baselines, and reproducible methodologies. This has led to contradictory findings across studies, with the best scorer in one study ranking among the worst in another. In this work, we introduce a benchmark for transferability scorers under a standardized evaluation protocol. As a baseline, our benchmark uses the model's accuracy on its source task (*e.g.*, ImageNet), which surprisingly outperforms most existing scorers. We then conduct a large-scale empirical study, evaluating 13 scorers across 10 vision model architectures and 11 datasets. Finally, we propose a novel combined scorer that leverages information from multiple individual scorers and demonstrate that it consistently outperforms all of them and the baseline.

## 1 Introduction

Deep learning practitioners today face an overwhelming choice: with hundreds of pre-trained models available through large open-source repositories such as Torchvision[1] and Hugging Face[2], how can one efficiently identify the model that will maximize performance for a specific task? Transfer learning offers a solution by leveraging knowledge acquired on source tasks to solve new target problems, but this abundance of choices creates its own challenges. The standard transfer learning approach (Kornblith et al., 2019; Zamir et al., 2018; Zhuang et al., 2020) involves pre-training on large-scale datasets (*e.g.*, ImageNet (Deng et al., 2009)) followed by fine-tuning on target tasks such as PASCAL VOC (Everingham et al., 2007) and SUN397 (Xiao et al., 2010). However, selecting the optimal model requires expensive fine-tuning and hyperparameter optimization across multiple pre-trained candidates to select the one with better performance.

The challenge of selecting the source model is compounded by the inherent difficulty of benchmarking transfer learning. In recent years, researchers have made efforts to establish comprehensive and fair evaluation protocols that account for various factors that affect transferability, such as source–target domain similarity, task type, and dataset size (Mensink et al., 2021; Dumoulin et al., 2021; Hosseinzadeh Taher et al., 2021). This complexity has motivated the creation of dedicated workshops, specifically focused on transfer learning challenges (Koch et al., 2024; Albalak et al., 2022), underscoring the importance of the field. More broadly, the science of benchmarking in machine learning has itself become a growing area of study (Hardt, 2025), further highlighting the need for rigorous evaluation protocols, a need that extends to transferability scoring. The research community has also produced extensive benchmarks and methodological examinations covering fine-tuning (Wortsman et al., 2022), adaptation challenges (Peng et al., 2017), open-set/few-shot scenarios (Dumoulin et al., 2021), and evaluation protocol design (Dumoulin et al., 2021).

Although these initiatives have significantly advanced the field, they also underscore the resource-intensive nature of empirical benchmarking, particularly when selecting the top-performing source models from large

---

[1] https://github.com/pytorch/vision
[2] https://huggingface.co/models

repositories. Transferability estimation (You et al., 2021; Agostinelli et al., 2022; Wang et al., 2023) emerged as a solution to identify promising source models without the computational overhead of fine-tuning. Given a target dataset and a collection of pre-trained models, transferability scorers estimate how well a single model will transfer to that target dataset. As a result, this approach reduces the traditional empirical comparison of models to a small subset of the most promising transfer candidates. However, comparing different transferability scorers has become challenging as there is no agreed-upon experimental design for transferability scorer evaluation across papers. This lack of standards leads to contradictory results and reproducibility issues. For example, the best-performing scorer in Pándy et al. (2022), LEEP Nguyen et al. (2020), appears among the worst scorers in Agostinelli et al. (2022).

In this work, we aim to *address the lack of standards in benchmarking transferability estimation scorers for image classification tasks and propose a systematic benchmark for future research. We focus on image classification using models pre-trained on ImageNet, which is arguably the most widely studied domain in the field of transferability estimation.* We focus on image classification because most existing transferability scorers target this task, and expanding to other tasks requires redesigning each method. Our first contribution is finding that the transferability literature lacks a robust baseline scorer, which limits progress monitoring as the literature advances. Prior work in transfer learning showed that the ImageNet baseline strongly correlates with downstream performance. (Kornblith et al., 2019). Despite its widespread use and established relevance in transfer learning field, the transferability estimation community has paid little attention to this measure in existing benchmarks. Early studies on transferability scorer's design (Bao et al., 2019; You et al., 2021; Xu and Kang, 2023; Li et al., 2023) perform only inter-scorer performance comparisons without establishing a baseline scorer. On top of that, Agostinelli et al. (2022) adopted the number of classes as the baseline, assuming that transferability can be trivially explained by the number of classes in a target dataset. However, several factors impact transferability, and no known linear relationship exists between the number of classes and transfer performance (Mensink et al., 2021). Given this limitation, we systematically adopt the source model's ImageNet accuracy as a baseline. Then, we empirically demonstrate and evaluate the capability of linear models to combine information from multiple scorers, creating a scorer with improved transferability predictions. Although simple, they provide an efficient way to perform such a combination. Our approach is flexible and can accommodate multiple transferability scorers, demonstrating superior performance compared to the best-performing individual scorers, including the ImageNet baseline.

Figure 1 illustrates our benchmark construction pipeline, the combined scorer and combined analysis. In particular, we present the following contributions:

- **Benchmarking of transferability scorers.** We present a systematic, large-scale benchmark for evaluating transferability scorers and validate it across 10 vision model architectures, 13 transferability scorers, and 11 datasets. We show that the performance of pre-trained models on ImageNet (before fine-tuning) is a simple and robust baseline, surprisingly outperforming most of the evaluated scorers. This baseline is often used in transfer learning literature, yet its transferability has remained largely unexplored until this work.
- **Combined analysis.** We propose a novel evaluation metric based on Kendall's Tau correlation that aggregates statistics across multiple datasets as an alternative to the traditional scorer-dataset basis. Unlike existing approaches, our metric provides a unified framework for comparing scorer performance across all datasets simultaneously.
- **Combined scorer.** We demonstrate the potential of using linear models to aggregate information from multiple base scorers by leveraging insights from our benchmark and combined analysis. Instead of relying solely on individual scorers, we evaluate the ability of linear models to combine multiple scorers and show that they outperform both the ImageNet baseline and the single best-performing scorers.

## 2    Related Work

Given the advantages of transferability estimation, it is not surprising that the literature is advancing rapidly, with an increasing number of transferability scorers being proposed. The literature has expanded from image classification (Nguyen et al., 2020; You et al., 2021; Wang et al., 2023; Xu and Kang, 2023) to

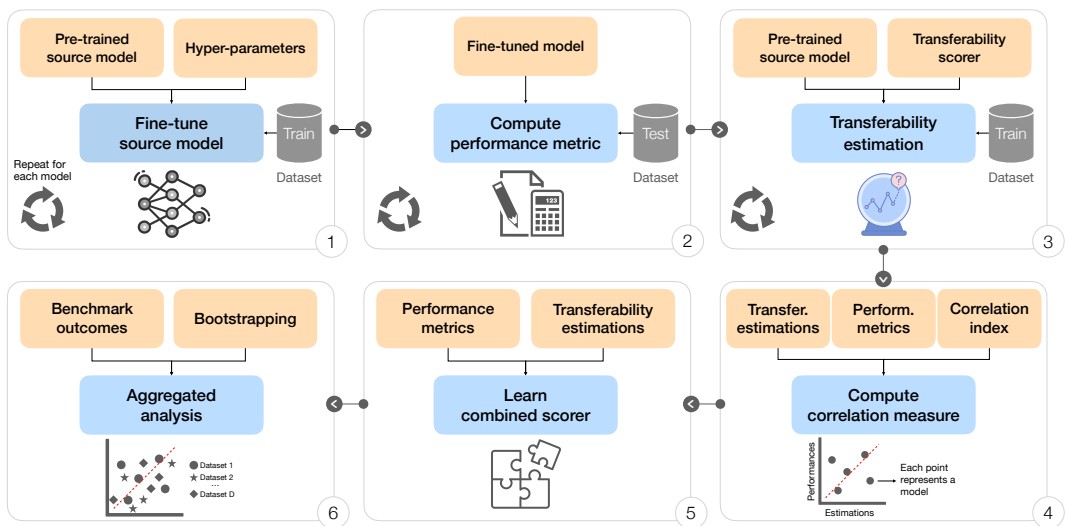

Figure 1: Our benchmark assesses transferability scorers through a multi-step evaluation. First, we individually fine-tune each pre-trained source model on a target dataset. For each fine-tuned model, we then compute both its actual performance metrics and its predicted transferability score. We repeat this process across all models to gather sufficient data for assessing the scorer's predictive quality. We then leverage this data to learn a combined scoring model. As the final step, we compute our proposed aggregated analysis, comparing the combined scorer's performance against the individual ones.

tasks as diverse as image segmentation (Pándy et al., 2022), text-based applications (Bai et al., 2023), and image retrieval (Dai et al., 2024). That said, we must clarify that the meaning of "transferability" across literature comprises at least: *task transferability*, which measures the relatedness of tasks (*e.g.*, classification, segmentation) under the same data (Zamir et al., 2018; Dwivedi and Roig, 2019); *dataset transferability*, which concerns the selection of the best datasets under the same model architecture (Achille et al., 2019; Vu et al., 2020; Gao and Chaudhari, 2021; Kim et al., 2023; Du et al., 2024); *architecture transferability*, which concerns the choice of different model architectures under the same pre-training source dataset (Nguyen et al., 2020; Tran et al., 2019; Bao et al., 2019; Ibrahim et al., 2022; Li et al., 2021; Bolya et al., 2021; Shao et al., 2022; You et al., 2021; Pándy et al., 2022; Ding et al., 2022; Gholami et al., 2023; Wang et al., 2023; Xu and Kang, 2023; Huang et al., 2022; Tan et al., 2021; Abou Baker and Handmann, 2024); and *checkpoint transferability*, in which both model architecture and source dataset may vary (Li et al., 2021).

Our scope focuses on architecture transferability, given the large number of open-source model architectures pre-trained on the same source dataset available on large model hubs, such as Torchvision or Hugging Face [3]. For instance, as of March 2026, Hugging Face and Torchvision comprise more than 25k image classification models. Beyond that, most practitioners rely on the traditional scheme of pre-training a source model on a large source dataset (*e.g.*, ImageNet) and then fine-tuning on the target dataset. Below we review prior work on architecture transferability. We organize existing methods into two main families based on the information they exploit from the source model. We focus on label-based approaches that operate on predicted class probabilities and feature-based approaches that leverage intermediate representations along with hybrid methods that combine both.

**Label-based.** LEEP (Nguyen et al., 2020) simulated an idealized classifier on probabilities estimated from the source and target class labels by taking the average log-likelihood of the expected empirical predictor. This simple classifier makes predictions based on the expected empirical conditional distribution between source and target labels. NCE (Tran et al., 2019) estimated the conditional entropy between the source and target labels. The limitations of such a label-based technique motivated practitioners to rely on features extracted, rather than requiring access to the source model's classifier.

---

[3] https://huggingface.co/models?pipeline_tag=image-classification

**Feature-based.** NLEEP (Li et al., 2021) adapted the probabilistic model of LEEP to exploit a Gaussian mixture model learned on such features. SFDA (Shao et al., 2022) uses the features to learn a Fisher Discriminant Analysis, robustified with hard examples, noise augmentation, and a class-separability criterion. LogME (You et al., 2021) learned a Bayesian linear classifier and uses the logarithm of its maximum evidence as the transfer score. PARC (Bolya et al., 2021), inspired by earlier works that used a "probe" model pre-trained on the target data, such as Dwivedi and Roig (2019); Dwivedi et al. (2020), avoided the choice (and training expense) of the probe by simulating it with a fixed embedding function on the ground-truth labels.

H-Score (Bao et al., 2019) measured the intra- *vs.* inter-class variance of features, while Regularized H-Score (referred to as R.H-Score in the following sections) (Ibrahim et al., 2022) added a shrinkage regularization term to improve feature's variance estimation. GBC (Pándy et al., 2022) estimates the feature overlap using the Gaussian Bhattacharyya Coefficient. TMI (Xu and Kang, 2023) takes a different approach than the above methods by proposing class-conditional entropy as a proxy for model adaptability, rather than relying on the (explicit or implied) preference for low intra-class variance.

PACTran (Ding et al., 2022) proposed a scorer based on the estimated PAC-Bayesian bound (McAllester, 1998) on the generalization error in a transfer-learning setting, using cross-entropy loss and considering Gaussian, Dirichlet, and Gamma as prior distributions. ETran (Gholami et al., 2023) is one of the few hybrid methods in the literature. It uses a simplified simulated classifier based on Linear Discriminant Analysis to evaluate the compatibility of the source model features with the target dataset classes. ETran also computes an energy/entropy-inspired evaluation of the separability of the feature space. The final score is a simple sum of the two components. NCTI (Wang et al., 2023) is another hybrid scorer that takes advantage of the empirical observation of *neural collapse* (Papyan et al., 2020), which predicts that deep models reach a specific geometric configuration when fully trained. The final score is the sum of three terms, each measuring a factor that composes the neural collapse property.

**Benchmarks.** Despite the growing importance of transferability scorers, comprehensive evaluation frameworks remain limited. Agostinelli et al. (2022) conducted an early evaluation across 5 transferability scorers and 9 target datasets. Our work significantly expands this evaluation scope, encompassing 13 transferability scorers and introducing a strong baseline method. We evaluate performance across 11 datasets spanning diverse task types (generalist, fine-grained classification, and medical imaging), providing a comprehensive assessment of scorer generalizability. Recognizing that benchmarking results can be highly sensitive to hyperparameter choices (as demonstrated in Section 3), we conduct an exhaustive exploration of 75 hyperparameter combinations to ensure optimal performance across all evaluated methods. In contrast, prior work evaluated only 6 combinations (Agostinelli et al., 2022).

**ImageNet performance as transferability scorer.** Kornblith et al. (2019) found a strong positive correlation between source model architectures' (*e.g.* ResNet50, EfficientNet) top-1 accuracy on ImageNet pre-training and their performance on several target datasets. Ericsson et al. (2021) confirmed these findings for self-supervised pre-trained models on few-shot learning and object detection downstream tasks. However, LEEP (Nguyen et al., 2020) is the only scorer we found that explicitly compared against ImageNet accuracy as a baseline, and only for a single dataset, which is absent in subsequent studies.

**Other scenarios.** For readers interested in transfer learning for robotics, we refer to the survey of Zhu et al. (2023). For transferability estimation beyond this scope, we point them to recent work on natural language processing (Garbaciauskas et al., 2024; Cho et al., 2024; Menta et al., 2024; Sun et al., 2024), multimodal models (Zhang et al., 2023; Dai et al., 2024; Shao et al., 2024), object detection (Wang et al., 2024), and meta-learning (Arango et al., 2024). Transferability scorers for foundation models introduce new challenges, such as the need for a per-modality scorer (Zohar et al., 2023), and revisiting what "transferability even means across heterogeneous pre-training sources and objectives.

## 3 Benchmark Design

Before describing our benchmarking design, we first provide a formal description of the problem at hand. Our goal is to estimate the transferability score of a collection of source model architectures for a particular target dataset. The target dataset consists of labeled images with corresponding ground-truth labels. The

defining criterion for a useful transferability scorer is that its estimated scores correlate strongly with the actual performance metric (e.g., accuracy) achieved by different model architectures on the target dataset. This ground-truth performance is measured by evaluating fine-tuned models on a test subset that is withheld from both the training and hyperparameter selection stages, ensuring an unbiased estimate of generalization. Since full fine-tuning is computationally expensive for large model collections, we seek to predict transfer performance using computationally efficient transferability estimation methods that avoid the costs of actual fine-tuning.

Despite this seemingly straightforward evaluation protocol, we found significant inconsistencies in experimental design throughout the literature. Previous work by Nguyen et al. (2020) fails to explicitly describe its experimental setup or provide details about the fine-tuning process. More concerning, some studies, such as (Shao et al., 2022; Wang et al., 2023; Xu and Kang, 2023), concatenate training and validation sets when fine-tuning models and computing the final performance metrics. These methodological choices lead to inconsistent performance of the transferability scorer and contradictory results in papers (Agostinelli et al., 2022; Pándy et al., 2022; Wang et al., 2023). To ensure transparent and reproducible results, we strongly recommend clearly documenting the complete evaluation procedure for transferability scorers. This work follows the standardized steps outlined as follows:

1. **Choose a target task** $T$ to fine-tune.
2. **Choose pre-trained model architectures** $\{A_1, A_2, \ldots, A_n\}$ for evaluation.
3. **Compute transferability scores** $S(M, T, A_i)$ for each scoring method $M$ and architecture $A_i$ using only the target training set.
4. **Obtain ground-truth performance** for each architecture $A_i$ by: (a) optimizing hyperparameters on the validation set to yield $A_i(T)$, then (b) evaluating the performance metric $P(T, A_i)$ on the test set.
5. **Evaluate scorer quality** by computing the correlation coefficient (*e.g.*, Kendall's $\tau$, Pearson's $r$) between the scores $S(M, T, A_i)$ and empirical performance metrics $P(T, A_i{}^{opt})$ across all architectures.

To compute the correlation coefficient described in Step 5, one must consider two cases: the transferability scores and the final model's performance metric on the target test set after fine-tuning. The latter's accuracy is related to the practitioner's experimental design, encompassing critical decisions regarding architecture selection, hyperparameter configurations, and optimization strategies. By using this evaluation framework, practitioners can systematically assess the performance of existing transferability scorers.

**Quality evaluation of traditional scorer.** The outcome/response in our design is the agreement or correlation between the transferability scores and the performance metrics across model architectures for each scorer. The current literature favored *weighted Kendall's tau* as a standard agreement measure (Appendix K). Traditional *Kendall's tau* correlation accounts for any monotonic correlation by considering all pairs of the two variables and counting agreements and disagreements (Vigna, 2015), while its weighted versions penalize disagreements among top-ranked elements. The weighted tau is defined as follows:

$$\tau_w(\mathbf{x}, \mathbf{y}) = \frac{\sum_{i<j} w(i, j) \, \text{sgn}(x_i - x_j) \, \text{sgn}(y_i - y_j)}{\sum_{i<j} w(i, j)}, \tag{1}$$

$$w(i, j) = v(\rho_{\mathbf{x}}(x_i)) + v(\rho_{\mathbf{x}}(x_j)) + v(\rho_{\mathbf{y}}(y_i)) + v(\rho_{\mathbf{y}}(y_j)), \tag{2}$$

$$v(r) = 1/(1 + r), \tag{3}$$

where $\mathbf{x} = (x_1, \ldots, x_n)$ contains the estimated transferability for a score, and $\mathbf{y} = (y_1, \ldots, y_n)$ contains their corresponding fine-tuning performance values, both across all architectures. Indices $i$ and $j$ denotes pairs of model architectures. The function $\rho_{\mathbf{z}}(z_i)$ denotes the rank of $z_i$ within the sample $\mathbf{z}$. Setting $w(i, j) = 1$ recovers the usual Kendall's tau.

**Combined analysis to assess the quality of the scorer's predictions.** Traditional experimental analysis is often limited to calculating a correlation measure for each target dataset. While this provides

insights into specific datasets, it becomes difficult to interpret as the number of datasets and scorers scales. This challenge is compounded by the lack of a single, comprehensive statistic that summarizes a scorer's overall performance across all datasets.

A naive approach to create such statistics is to pool all performance metrics and transferability estimations across all datasets without making statistical adjustments. Correlation measures such as the traditional tau statistics above (Equation 1) often assume that all data pairs are comparable. It holds within a single dataset in which measures share a similar scale. However, it breaks down when combining data from multiple datasets with vastly different performance ranges and estimation magnitudes. This problem manifests as distinct clustering patterns in scatter plots, as observed in Figure 5 of Tan et al. (2021) and Agostinelli et al. (2022), where data points from different target datasets form separate clusters.

We instead propose the **aggregated weighted tau**, which adapts the tau statistic to consider only the pairs within datasets. Our idea is to create a summarized statistics based on within-dataset measures. Considering a list of $N$ groups of data $\mathbf{G} = \{(\mathbf{x}_1, \mathbf{y}_1), \cdots, (\mathbf{x}_N, \mathbf{y}_N)\}$, where each pair $(\mathbf{x}_i, \mathbf{y}_i)$ is a complete set of samples $\{(x_j, y_j) \in (\mathbf{x}_i, \mathbf{y}_i)\}$. Then, the aggregated weighted tau is defined as:

$$\mathring{\tau}_{\mathrm{w}}(\mathbf{G}) = \frac{\sum_{(\mathbf{x},\mathbf{y}) \in \mathbf{G}} \tau_{\mathrm{num}}(\mathbf{x}, \mathbf{y})}{\sum_{(\mathbf{x},\mathbf{y}) \in \mathbf{G}} \tau_{\mathrm{den}}(\mathbf{x}, \mathbf{y})}, \tag{4}$$

where $\tau_{\mathrm{num}}(\mathbf{x}, \mathbf{y}) = \sum_{i<j} w(i,j) \operatorname{sgn}(x_i - x_j) \operatorname{sgn}(y_i - y_j)$, and $\tau_{\mathrm{den}}(\mathbf{x}, \mathbf{y}) = \sum_{i<j} w(i,j)$.

The variables $x_i$ and $y_i$ are the scorer's estimations, and performance metrics, respectively. We present some numerical examples of how the proposed weighted tau compares to traditional tau when data comes from different groups in Appendix D, and proofs about how it accounts for within-group statistics (Theorem 1), and bounds (Theorem 2) in Appendix E.

**Defining baselines for transferability scorers.** To date, the transferability estimation literature lacks baseline scorers. The findings of Kornblith et al. (2019) showed a strong linear relationship between transfer performance and ImageNet top-1 accuracy across several neural networks and datasets. To our knowledge, no other strong indicator of fine-tuning performance exists, such as ImageNet top-1 accuracy. Thus, that estimation meets our expectations from a robust baseline, *i.e.*, it is (i) easy and cheap to compute; (ii) has a simple interpretation; (iii) and a minimal proof of efficacy. While this relationship is well known in the transfer learning literature (Kornblith et al., 2019; Recht et al., 2019), it remains largely overlooked in transferability estimation research. That said, we adopt ImageNet top-1 accuracy as a baseline scorer for the rest of this paper.

## 4 Combined Transferability Scorer

Our benchmark produces a large number of performance data points for each scorer across multiple architectures and datasets. While prior works in transferability scorer design used this information primarily for comparative reporting, we recognize an opportunity to transform these individual performances into a robust transferability estimator. Instead of treating the diverse scorer outputs as competing alternatives, we use them as complementary sources of information. Individual transferability scorers capture different theoretical and empirical aspects of transfer learning. Some focus on feature separability (H-Score), others on probabilistic alignment (LEEP), and others on energy-based measures (ETran). Rather than selecting a single perspective, we propose combining these complementary views through learning into a combined scorer. We empirically adopt linear models for performing such combinations, as they provide interpretable weights that reveal each scorer's contribution.

### 4.1 Combination Strategies

To create an enhanced combined scorer, we explore strategies for combining the predictive signals of individual scorers. Our central hypothesis is that a weighted combination of scorers can outperform any individual scorer by leveraging their complementary strengths. The key design question is: *what sources of variation should the combination account for?*

Guided by this question, we build a progression of strategies, from coarse to fine, that account for increasingly detailed structure in the data. We begin with univariate approaches that calibrate each scorer independently, then move to multivariate approaches that learn joint relationships across scorers.

**Notation.** Let $\{(m_i, t_i, d_i, s_i, a_i)\}_{i=1}^N$ denote $N$ calibration and training tuples, where $m_i \in \mathbb{R}$ is the measured performance metric and $t_i \in \mathbb{R}$ is the transferability score. The categorical variables $d_i \in \mathcal{D}$, $s_i \in \mathcal{S}$, and $a_i \in \mathcal{A}$ indicate the target dataset, scorer, and model architecture, respectively. We define the quantities of unique elements as $D = |\mathcal{D}|$, $S = |\mathcal{S}|$, and $A = |\mathcal{A}|$. We apply Z-normalization (subtracting the group mean and dividing by the group standard deviation) per $(d, s)$ tuple groups to ensure comparability across heterogeneous scorers and datasets.

## 4.2 Training and Evaluation Protocol

We evaluate each combination strategy using a leave-one-dataset-out scheme (Arlot and Celisse, 2010) that rigorously tests generalization to unseen domains. This ensures the combination strategies perform effectively in realistic scenarios where practitioners need to estimate transferability for new target datasets.

The evaluation process works as follows: to predict performance on a target dataset $d_{target} \in \mathcal{D}$, we train a combination strategy on the z-normalized tuples from the $D-1$ complementary datasets $D_{train} = \{d_i \in \mathcal{D} \mid d_i \neq d_{target}\}$. This training subset provides $N_{train} = A \times |D_{train}|$ observations per scorer. After fitting the combination strategies on $D_{train}$, we perform inference on $d_{target}$ to output a predicted performance vector $\hat{\mathbf{m}} \in \mathbb{R}^A$ corresponding to the $A$ available architectures.

## 4.3 Univariate Strategies

Different transferability scorers operate on different scales. A scorer's raw output may correlate well with transfer performance yet require calibration before it can be compared or combined with others. In the univariate strategies, we explore differences in *how much structure* they use when learning this calibration.

**Ignoring all structure (Overall).** The simplest approach pools all $A \times |D_{train}| \times S$ available training tuples into a singular set, ignoring which scorer or dataset produced each observation. We fit a single linear regression using transferability scores ($t_i$) to predict aligned performance metrics ($m_i$): $m_i = \beta_0 + \beta_1 t_i$, where $\beta_0, \beta_1$ are the global regression parameters. At inference, we evaluate $s_{base} \in \mathcal{S}$ on the target dataset: $\hat{m}_a = \beta_0 + \beta_1 t_{\text{test},a}^{s_{base}}$. This outputs the prediction vector $\hat{\mathbf{m}} \in \mathbb{R}^A$. This approach assumes all scorers share a common functional relationship with performance while relying on a singular raw proxy during inference.

**Accounting for scorer heterogeneity (Per Scorer).** Different scorers produce outputs on varying scales and may exhibit specific biases. To address this, we train a separate linear model for each scorer $s \in \mathcal{S}$ exclusively on its $A \times |D_{train}|$ training tuples: $m_i^s = \beta_0^s + \beta_1^s t_i^s$. At inference on $d_{target}$, we evaluate each scorer's model using its distinct test scores $t_{\text{test},a}^s$. Next, we average the outputs across all scorers for each architecture: $\hat{m}_a = \frac{1}{S} \sum_{s \in \mathcal{S}} (\beta_0^s + \beta_1^s t_{\text{test},a}^s)$, resulting in a prediction vector $\hat{\mathbf{m}} \in \mathbb{R}^A$. This accounts for scale differences but assumes each scorer's relationship with performance is consistent across datasets.

**Accounting for scorer–dataset interactions (Per Scorer, Dataset).** To capture finer-grained variations, we partition the calibration data by both scorer and dataset, training a single regression model for each $(s, d)$ pair using its $A$ training tuples: $m_i^{s,d} = \beta_0^{s,d} + \beta_1^{s,d} t_i^s$. At inference on $d_{target}$, we evaluate every combination: for each scorer $s \in \mathcal{S}$ measured on test inputs $t_{\text{test},a}^s$, we produce inferences leveraging all $|D_{train}|$ dataset relationships learned for that scorer, yielding $S \times |D_{train}|$ distinct prediction. The final prediction vector $\hat{\mathbf{m}} \in \mathbb{R}^A$ is the average across all models: $\hat{m}_a = \frac{1}{S \times |D_{train}|} \sum_{s \in \mathcal{S}} \sum_{d \in D_{train}} \left( \beta_0^{s,d} + \beta_1^{s,d} t_{\text{test},a}^s \right)$.

## 4.4 Multivariate Strategies

The univariate strategies described above calibrate each scorer independently before combining their predictions. However, the scorers may provide complementary information, so errors made by one scorer could be corrected using the signal from another. In contrast, multivariate strategies treat all scorer outputs as features simultaneously, learning to weight their combined and complementary contributions.

**Least Squares Regression. Least Squares Regression.** We build a feature matrix $\mathbf{X}_{train} \in \mathbb{R}^{N_{train} \times S}$, where rows correspond to the $N_{train} = A \times |D_{train}|$, and $\mathbf{Y}_{train} \in \mathbb{R}^{N_{train}}$ is the ground-truth architecture metrics. We learn a coefficient vector $\boldsymbol{\beta} \in \mathbb{R}^S$ and a scalar intercept $b$ minimizing the reconstruction error: $\min_{\boldsymbol{\beta},b} \|\mathbf{Y}_{train} - (\mathbf{X}_{train}\boldsymbol{\beta} + b)\|_2^2 \implies \hat{\boldsymbol{\beta}} = (\mathbf{X}_{train}^T \mathbf{X}_{train})^{-1} \mathbf{X}_{train}^T \mathbf{Y}_{train}$. Final performances approximations are defined as $\hat{\mathbf{Y}}_{target} = \mathbf{X}_{test}\hat{\boldsymbol{\beta}} + b \in \mathbb{R}^A$.

**Support Vector Regression.** The approaches above assume linear relationships between scorer outputs and performance. To capture potential non-linear configurations, we use a Support Vector Regressor with a gaussian (RBF) kernel $\phi(\cdot)$. We use the matrices $\mathbf{X}_{train} \in \mathbb{R}^{N_{train} \times S}$, $\mathbf{Y}_{train} \in \mathbb{R}^{N_{train}}$, we implement: $\min_{\mathbf{w},b} \frac{1}{2}\|\mathbf{w}\|_2^2 + \alpha \sum_{i=1}^{N_{train}} \max(0, |\mathbf{Y}_{train}^{(i)} - (\mathbf{w}^T\phi(\mathbf{X}_{train}^{(i)}) + b)| - \varepsilon)$, where the conditional $\varepsilon$, and $\alpha$ controls the fitting. In inference we adopt $\hat{\mathbf{Y}}_{target} = \mathbf{w}^T\phi(\mathbf{X}_{test}) + b \in \mathbb{R}^A$.

## 5 Experiments

**Model Architectures.** All methods predict transferability for architecture-level image classification, so we restrict both source and target tasks to this domain. Our architecture pool comprises 10 ImageNet-pretrained architectures spanning five families: ResNet-18,34,50 (He et al., 2016), DenseNet-121,161,169 (Huang et al., 2017), MobileNetV2-0.5,1.0 (Sandler et al., 2018), EfficientNet-B0 (Tan and Le, 2019), and ViT-small (Dosovitskiy et al., 2020). We take model's implementations and checkpoints from TIMM (Wightman, 2019).

**Datasets.** To assess performance across varying difficulty levels, we fine-tuned these architectures on 11 datasets grouped by their distance from ImageNet: "generalist" (Caltech-101 (Fei-Fei et al., 2004), SUN397 (Xiao et al., 2010), PASCAL VOC2007 (Everingham et al., 2007)), "fine-grained/natural" (Oxford Flowers 102 (Nilsback and Zisserman, 2008), Oxford-IIIT Pets (Parkhi et al., 2012)), "fine-grained/artifacts" (FGVC-Aircraft (Maji et al., 2013), DTD (Cimpoi et al., 2014), Stanford Cars (Krause et al., 2013)), and "fine-grained/medical" (BrainTumor (Cheng et al., 2015), BreakHis (Spanhol et al., 2016), ISIC19 (Codella et al., 2018)). Those datasets cover a wide range of classes (2–397), training samples (2 040–19 850), difficulty, and imbalanced cases (all medical datasets). More information about each dataset appears in Appendix G.

**Fine-tuning protocol.** It follows the Tuning Playbook guidelines (Godbole et al., 2023) for each model. We use the Halton quasi-random sequences (Halton, 1964) to explore 75 hyperparameter combinations of learning rate and weight decay (You et al., 2021), simulating a realistic scenario in which practitioners invest substantial effort to produce high-quality fine-tuned models. We use SGD with Nesterov momentum and cosine learning rate scheduling, training for 100 epochs. Hyperparameter selection occurs on a validation split to ensure fair evaluation. We report results for the full-model fine-tuning scenario, as this approach is common in practice and presents the most demanding evaluation for transferability scorers. Frozen-feature results appear in Figures 5, 6, and 7 in Appendix C. We evaluate transferability using both the weighted tau statistic and our proposed aggregated weighted tau across combined experiments.

**Transferability Scorers.** We consider 13 state-of-the-art transferability scorers by choosing each implementation with available source code or reproducing it ourselves. For scorers with multiple versions or tunable hyperparameters, we adopted the configuration reported as optimal in the original publication. We summarize each scorer in Appendix K.

**Evaluation protocol.** We adopt our proposed benchmark design (Section 3). We adopt the balanced accuracy as the standard performance metric for all datasets. For baseline purposes, we use each model's ImageNet top-1 accuracy and measure its correlation with the model's fine-tuned performance metric.

Assessing the quality of a scorer involves calculating a correlation metric across multiple pre-trained models using a specific dataset. Assessing scorer quality requires estimating a correlation over a finite set of pretrained architectures. To quantify the stability of that estimate, we apply bootstrap resampling to the paired observations for each architecture. At each bootstrap iteration, we sample the architectures with replacement, recompute the correlation statistic on that resampled set, and repeat this process many times. We then report the mean estimate of the correlation, along with an uncertainty measure derived from the bootstrap distribution. This procedure measures the robustness of the scorer–performance correlation

| | Cal-101 | SUN397 | VOC 2007 | Ox.Flowers | Ox.Pets | Aircraft | DTD | Stan.Cars | BrainTumor | BreakHis | ISIC 19 | Averaged |
|---|---|---|---|---|---|---|---|---|---|---|---|---|
| ImageNet | 76 | 80 | 74 | 64 | 81 | 76 | 66 | 64 | 13 | 47 | 53 | 63 |
| NCTI | 76 | 94 | 87 | 74 | 73 | 85 | 72 | 81 | 10 | 13 | 62 | 66 |
| ETran | 88 | 43 | 75 | 59 | 73 | 76 | 42 | 77 | 41 | 77 | 68 | 65 |
| PACTran | 29 | 14 | 33 | 56 | 37 | 60 | 51 | 75 | 54 | 7 | 8 | 38 |
| GBC | 67 | 79 | 73 | 53 | 47 | 90 | 59 | 44 | -14 | 8 | -4 | 46 |
| PARC | 80 | 78 | 55 | 51 | 39 | 77 | 70 | 39 | -36 | 23 | 9 | 44 |
| NLEEP | 59 | 82 | 72 | 51 | 62 | 20 | 48 | 67 | -30 | 38 | -17 | 41 |
| LEEP | 68 | 73 | 84 | 30 | 60 | 18 | 48 | 44 | -2 | -7 | 25 | 40 |
| LogME | 57 | 70 | 63 | 42 | 42 | 11 | 41 | 25 | -29 | 12 | 56 | 35 |
| TMI | 30 | 6 | 36 | 37 | 28 | 22 | 56 | 57 | 36 | 5 | 40 | 32 |
| NCE | 68 | 91 | 36 | -36 | 61 | 16 | 52 | 45 | -8 | -24 | -13 | 26 |
| SFDA | 6 | 49 | 35 | 54 | 21 | 36 | 23 | 49 | -2 | -0 | -3 | 24 |
| H-Score | 10 | 50 | 34 | -38 | 14 | -27 | 2 | 8 | -13 | 47 | 43 | 12 |
| R.H-Score | 11 | 46 | 15 | -9 | 13 | -31 | -7 | -11 | -49 | -35 | -25 | -7 |

Figure 2: Benchmark results. We report the weighted tau ($\tau_{\mathrm{w}} \times 100$), higher is better. Row groups, from top: ImageNet baseline, and state-of-the-art scorers performance. Dataset groups, from left: generalist, natural, artifacts, medical. Averaged: average of individual dataset correlation measures. The ridgeline plots show the distribution of 1000 bootstrap iterations, with the mean shown in the figures. The ImageNet scorer baseline (first row) outperforms most existing scorers, except for NCTI and ETran. The latter outperformed the baseline by a tight margin.

estimate. Although bootstrapping is a well-established technique in statistics, applying it to benchmark transferability and to quantify the reliability of scorer-correlation estimates is, to our knowledge, novel.

## 5.1 Benchmark Results

Our results appear in Figure 2. Each cell shows the distribution of the correlation measures after 1000 bootstrapping iterations and the average value of those samples (in numbers). The distribution refers to the scorer's correlation index, *i.e.*, how the estimations correlate with the final performance when using that scorer (row) to predict transfer performance in a particular dataset (column). The four dataset groups appear in columns, representing the target domain distance, from left to right: generalist, fine-grained/natural, fine-grained/artifacts, and fine-grained/medical. All correlation measures consider the traditional weighted tau, reported as $\tau_w \times 100$ for readability.

The isolated row at the top shows ImageNet top-1 performance, used as a baseline. The second-row group (NCTI to R.H-Score) shows the performance of state-of-the-art scorers. Mean values and bootstrapping distribution shapes vary wildly across dataset groups and scorers for both standard computer vision datasets and medical ones. All scorers failed to predict the transfer in the latter scenario, indicating consistent difficulty across all scorers, with the ETran scorer being the lone exception.

The Averaged column shows the mean performance for each scorer. Only NCTI and ETran scorers outperformed the ImageNet baseline by a tight margin. All scorers below ETran showed unstable and unreliable performance, with an average correlation below 50. Some recorded negative values, which is contrary to the expected design of transferability scorers.

## 5.2 Combined Analysis

Figure 3 shows the global scorer's trend across all datasets and architectures (110 points in our case). The scorer's name appears at the bottom right, and the proposed aggregated tau ($\mathring{\tau}_{\mathrm{w}}$) appears on top left. We use a rank-rank plot to display all results on a common scale, where ranks are computed per dataset. A better scorer exhibits more points concentrated at the main diagonal since the ranks of the scorer's estimations match the ranks of fine-tuned performances. Our proposed aggregated analysis provides a global overview of the scorer's trends and visualization scales as the dataset and the number of scorers increase. Observe

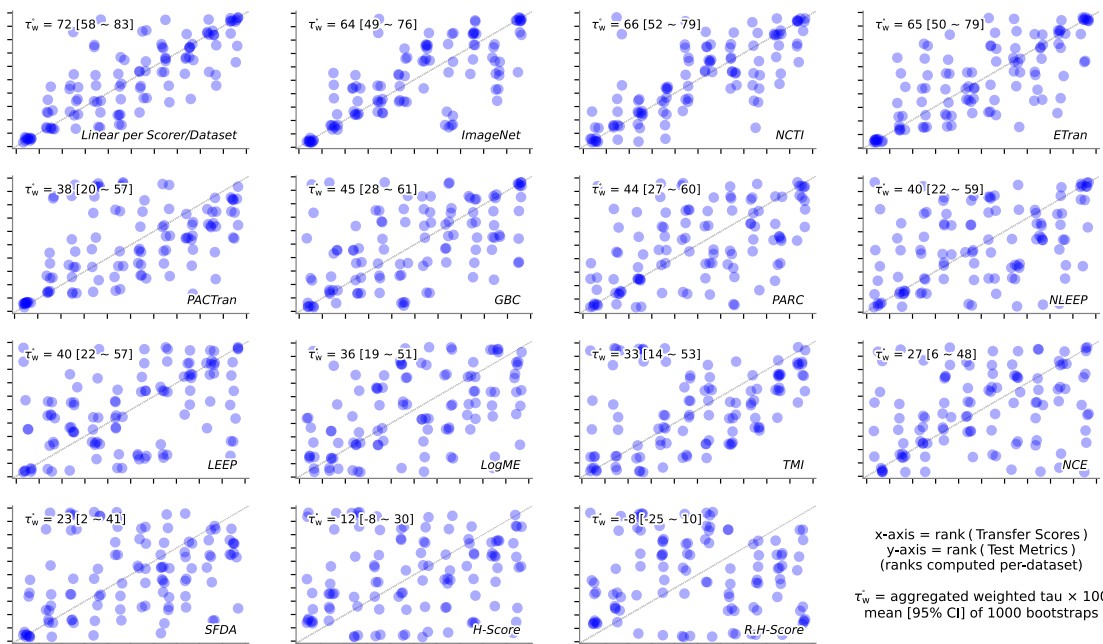

Figure 3: The plots illustrate our proposed combined analysis for all datasets within our benchmark. We show the results for the best combination strategy that yields the best combined scorer (Section 5.3), ImageNet baseline, and all state-of-the-art scorers. The combined analysis is possible due to the proposed aggregated tau (Section 3), and bootstrapping is employed to compute its 95%-confidence intervals, shown in brackets on each plot. The main message of those plots is whether the data points concentrate at the main diagonal of the plot, showing the scorer's ability to match the ranks of transfer scores and test performance metrics.

that the four worst scorers (NCE, SFDA, H-Score, and R.H-Score present scattered points far from the main diagonal, in which the best models present the best rank-rank matching close to the main diagonal. This indicates they commonly fail to predict transferability performance for a particular dataset or model.

From a practical perspective, our proposed aggregated tau statistic and visualizations provide a global trend in a scorer's performance and can accommodate any number of datasets. This feature addresses a limitation found in previous works, which do not provide a statistic that shows trends across many target datasets. Consequently, researchers are often required to conduct exhaustive correlation inspections for each dataset to determine the best scorer for their specific needs. Our proposed analysis enables practitioners to evaluate any number of scorers and identify the top performer using a single, unified metric.

### 5.3 Combined Scorer Performances

The combination strategies are versatile tools that explore available information across different datasets and scorers to produce a combined scorer. The effectiveness of any learning-based method is closely tied to the quality of the calibration tuples. In this context, we examine three distinct scenarios regarding the availability of quality information to train a combined scorer: 1) Top performers: only the highest-quality information from the best-performing scorers is used to create an improved scorer. 2) mid performers: Information from scorers with moderate performance. This scenario aims to understand how combination strategies behave with reasonably good but suboptimal data. 3) bottom performers: information from the lowest-performing scorers, testing under a challenging and noisy conditions.

We evaluate five methods to create the combined scorer: **Overall**, **Per Scorer**, **Per Scorer-Dataset**, **Least Sq**, and **SVM**. For each choice, we consider the scenarios of data availability: the three top scorers (NCTI, ETran, PACTran) with and without ImageNet; the three bottom scorers (SFDA, H-Score, R.H-Score) with and without ImageNet; and the 7 mid-range scorers. We present the results without ImageNet in the

calibration data in the main text (Figure 4), and those including ImageNet are provided in the appendix (Figure 11). Additionally, we examine how the number of available scorers affects average performance in Appendix H, along with their corresponding run times in Appendix I.

| | Cal-101 | SUN397 | VOC 2007 | Ox.Flowers | Ox.Pets | Aircraft | DTD | Stan.Cars | BrainTumor | BreakHis | ISIC 19 | Averaged | Combined |
|---|---|---|---|---|---|---|---|---|---|---|---|---|---|
| ImageNet | 76 | 80 | 74 | 64 | 81 | 76 | 66 | 64 | 13 | 47 | 53 | 63 | 63 |
| Overall (Top) | 29 | 15 | 32 | 57 | 38 | 60 | 48 | 75 | 53 | 6 | 8 | 38 | 38 |
| Per Scorer (Top) | 93 | 56 | 77 | 77 | 92 | 82 | 87 | 88 | 40 | 26 | 66 | 71 | 71 |
| Per Scorer,Dataset. (Top) | 93 | 57 | 77 | 76 | 91 | 81 | 88 | 89 | 41 | 27 | 67 | 72 | 72 |
| Least Sq. (Top) | 91 | 55 | 81 | 74 | 92 | 82 | 91 | 89 | 34 | 23 | 75 | 72 | 72 |
| SVM (Top) | 80 | 44 | 76 | 45 | 87 | 72 | 86 | 84 | 33 | 24 | 82 | 65 | 65 |
| Overall (Mid) | 71 | 91 | 39 | -35 | 61 | 18 | 52 | 46 | -8 | -25 | -16 | 27 | 26 |
| Per Scorer (Mid) | 88 | 86 | 92 | 78 | 61 | 74 | 84 | 43 | -17 | 6 | -6 | 54 | 53 |
| Per Scorer,Dataset (Mid) | 88 | 86 | 92 | 79 | 61 | 74 | 83 | 44 | -18 | 4 | -7 | 53 | 53 |
| Least Sq. (Mid) | 75 | 75 | 71 | 77 | 56 | 47 | 81 | 52 | 0 | 6 | 23 | 51 | 51 |
| SVM (Mid) | 58 | 39 | 54 | 65 | 61 | 77 | 58 | 55 | 20 | 28 | 43 | 51 | 51 |
| Overall (Bot) | 10 | 50 | 30 | -36 | 17 | -29 | 1 | 7 | -13 | 46 | 42 | 11 | 11 |
| Per Scorer (Bot) | 2 | 48 | 36 | 13 | 21 | 26 | 22 | 44 | -1 | 3 | -2 | 19 | 20 |
| Per Scorer,Dataset (Bot) | 4 | 50 | 34 | 13 | 22 | 25 | 24 | 46 | -0 | 5 | 1 | 20 | 20 |
| Least Sq. (Bot) | 1 | 33 | 3 | -9 | 15 | 39 | 29 | 56 | 23 | 39 | -3 | 21 | 20 |
| SVM (Bot) | 30 | 38 | 3 | 24 | 12 | 24 | 43 | 47 | 25 | -14 | 6 | 22 | 21 |

Figure 4: Ablation study on how training data quality affects the performance of each strategy to combine scorers. Each row evaluates a combination method across several target datasets (columns). To analyze the impact of data quality, we train each strategy on three subsets of the source data, indicated by the -Top, -Mid, and -Bot suffixes. These subsets contain data on scorers who performed best (top), average (mid), or worst (bottom) on our benchmark (Figure 2). We compare these results with the ImageNet baseline shown in the top row. The "Overall" and SVM strategies are consistently the worst for all cases. In general, univariate strategies, when combined with high-quality data, are sufficient to outperform the baseline by a large margin in both averaged and combined analyses. Section 5.3 provides a detailed analysis.

Combination strategies trained with top-performing scorers surpassed the ImageNet baseline's mean correlation, except for the "Overall" method, which is the simplest combination method form: a single regression across all available data, ignoring the influence of specific datasets or scorers. Per Scorer; Per Scorer, Dataset and Least Squares showed similar performance across all scenarios, with minor distinctions between individual, averaged, and combined distributions. When trained on data from Mid- and Bottom-tier scorers, the naive Overall strategy consistently performs worst. Under the same data conditions, other univariate (Per Scorer; Per Scorer-Dataset) and multivariate (Least Sq.; SVM) strategies also fail to outperform the ImageNet baseline, with their failures more evident on the medical datasets. This underscores the critical importance of training data quality for building a reliable combined scorer. Our results demonstrate that using data exclusively from top-performing scorers yields a superior estimator that not only surpasses other methods but also outperforms the strong ImageNet baseline, especially on these challenging medical datasets.

### 5.4  Findings

**Only two state-of-the-art scorers outperform the ImageNet as baseline.** Comparing the first row of Figure 2 to the SOTA results in the second row group reveals that ImageNet performance consistently outperforms most existing scorers. According to both individual and averaged correlation measures, only two scorers (NCTI and ETran) achieve competitive results with this baseline, underscoring the significance of this finding. Our analysis (Figures 2, and 3) indicates that practitioners can rely on source performance for informed decisions about final transfer performance, particularly in ImageNet-centric research on architecture transferability, as we expect any newly proposed transferability scorer must, at least, outperform this simple baseline. Moreover, the ImageNet performance of source models offers a practical, cost-free advantage, since researchers typically report these performance numbers in papers or open-source repositories.

**Averaged and combined taus have similar averages but different distributions.** In practice, the combined tau provides a global summary of a scorer's performance that is independent of the number of

datasets. As shown in Figure 3, this property allows practitioners to quickly assess overall scorer performance regardless of the number of datasets considered. Additional results in Appendix F show that combined taus produce more sharply peaked bootstrap distributions, indicating lower estimator uncertainty.

**Calibration data matters more than modeling complexity.** Our ablation results (Figure 4) reveal that calibration data dominates other design choices. The Overall strategy, which pools all data without accounting for scorer-specific scales, consistently performs worst regardless of input quality. In contrast, Per Scorer and Per Scorer, Dataset achieve nearly identical performance, suggesting that accounting for scorer heterogeneity is essential, but modeling scorer–dataset interactions provides limited benefit. Similarly, the multivariate strategies (Least Squares, SVM) offer no systematic advantage over the simpler Per Scorer approach. These findings suggest that practitioners can adopt the straightforward Per Scorer calibration without sacrificing performance. Most critically, the quality of input scorers matters more than the combination strategy. To support this claim, we conduct an ablation study considering three data availability scenarios: Top, Mid, and Bottom quality, as shown in Figure 4. Strategies leveraging Top-quality information consistently yield stronger models, with consistent performance degradation for Mid- and Bottom-quality scorers. The Overall scorer remains an outlier and continues to perform poorly across all scenarios.

**Domain distance between source and target datasets negatively affects the scorers.** The relationship between domain distance and scorer performance reveals a clear degradation pattern across dataset categories. We organized the columns in Figure 2 into four categories: generalist, fine-grained/natural, fine-grained/artifacts, and fine-grained/medical, respectively. As we examine these categories in order of increasing difficulty, the benchmark distribution means consistently decrease until they reach a critical threshold where nearly all scorers present mean values of 50 or below, indicating unreliable correlation.

**Medical datasets remain challenging for transferability estimation.** All scorers, including the ImageNet baseline, perform inconsistently, except for ETran. Unlike other methods, ETran explicitly measures domain shift between the source and target domains, which is likely critical given the large gap between ImageNet pretraining and medical imaging. Although ETran shows promising results, further analysis is necessary to determine whether explicitly incorporating domain shift into the final score is necessary for reliable transferability estimation in medical applications. In addition, models with similar architectures often achieve comparable performance on medical datasets (Table 1 in Appendix B). This pattern rarely appears in natural image benchmarks. We hypothesize that existing scorers lack the sensitivity needed to reliably rank models in this domain and capture minimal performance differences across model architectures.

## 6 Conclusion

The growing importance of transfer learning has sparked significant interest in estimating transferability, leading to the development of increasingly sophisticated theoretical and practical techniques. Our work advances the latter field in three main ways. First, we demonstrate how to create a combined scorer using linear models, with predictions from transferability metrics as calibration data. All combination strategies are fast to train (Appendix I) and incur a one-time cost, allowing practitioners to reuse the combined scorer extensively across unseen datasets. Second, we benchmark architecture transferability across 10 neural network architectures and 13 transferability scorers, using a standardized experimental design and an extensive hyperparameter search that simulates practitioner efforts to achieve the best final model. Finally, we propose a combined correlation index that better summarizes the global trends of each scorer and incorporates group-wise statistics between datasets. Note our proposed methodology (Section 3) is agnostic to the task and can also be used for foundational models when the literature intersect this model family.

Our experimental design includes four dataset categories to assess how the domain distance between source and target datasets affects the performance of the scorers, and the results showed that all scorers struggled with medical datasets. Our work provides a standardized evaluation protocol that accommodates many transfer learning scenarios and is suitable for future research in both transfer learning and transferability estimation. Additionally, our results indicate that even state-of-the-art scorers often fail to outperform a simple ImageNet baseline scorer. From a practical viewpoint, we expect all scorers to surpass this simple but challenging ImageNet baseline. Based on that, we recommend future studies on transferability estimation, benchmarking, and scorer design to include ImageNet as a baseline scorer.

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
