# OpenReview forum: "Benchmarking Transfer Learning: From Simple Baselines to Combined Scorers for Transferability Estimation"
_TMLR — Decision pending for TMLR_

### Review · Reviewer_D5Rq · 2026-05-08

**Summary Of Contributions:**

The authors address the issue of selecting the best pre-trained models. The authors note that there is currently a lack of standardized assessment protocols, uniform baseline measures, and reproducible methods in the field of transferability assessment. For this reason, they introduce a benchmark for transferability scorers under a standardized evaluation protocol. They conduct a large-scale empirical study, evaluating 13 scorers across 10 vision model architectures and 11 datasets. In addition, they propose a novel combined scorer that leverages information from multiple individual scorers and demonstrate that it consistently outperforms all of them and the baseline.

key strengths:
- The authors are conducting a comprehensive study and developing a transparent benchmark.
- Many experiments were performed on various networks and datasets.
- The authors provide good motivation and explain their approach.
- The authors present new methods, such as aggregated weighted tau and combination strategies for the scorers.
- The authors provide details on various points in the appendix, such as datasets, baseline methods, etc. This is very helpful for understanding the material.

key  weaknesses:
- In some places, the paper is not clearly worded or is slightly misleading.
- I am missing the connection to foundation models such as CLIP.

**Audience:**

Yes

**Audience Explanation:**

The paper provides a good overview of the research in this field, tests various baseline methods, and examines contradictory findings from previous papers. I think that it is not necessarily the most modern topic, but the results are interesting.

**Claims And Evidence:**

Yes

**Claims Explanation:**

- The authors state that they are presenting a transparent benchmark, supported by implementation details and code. In addition, they use freely available networks and datasets. The authors have addressed the issues with other benchmarks and strive to evaluate everything as fairly and transparently as possible.
- I like that they are also thinking about which scoring system is most appropriate in theory.


- The authors state that the assessment of transferability remains a highly researched and important topic. The selected baselines are from 2023 or earlier, and I find there is a lack of reference to foundation models, which perform well in zero-shot tasks across various datasets.
- The motivation behind aggregated weighted tau is explained, but I lack experiments to support it.

**Requested Changes:**

Requested Changes:
- I think that it is important to make connections to foundational models and explore new methods for assessing transferability (later 2023).
- The weighted tau metric is the only one used for the evaluations; I would like to see a comparison with traditional quality evaluation metrics to assess the added value.
- Sections 5.3 and A8 compare the various scorers, but I find the wording regarding “data” and “dataset” confusing, since the same datasets are being used.
- If I understand correctly (second paragraph in section 4.4), all but one of the datasets are used to train a combination strategy. I would like to see if all the experiments/datasets are necessary to achieve a good transfer score. It would also reduce the runtime.
- Add references to all sections of the appendix.

---

> ### Author Response · Authors · 2026-06-13
>
> We thank the reviewer for the thoughtful feedback and the recognition of our contributions. Below we address each point in turn.
>
> ---
>
>  ### **RC-1: Foundational models and exploring new methods for assessing transferability (later 2023).**
>
> We want to address a potential misunderstanding regarding our contribution by clarifying that the framing of "newer methods are missing" does not accurately reflect the central intent of our paper. Our primary observation is methodological: we find that the field currently lacks a standardized protocol, a robust baseline, and reproducible methodologies. This problem is agnostic to which scorers are in the pool. Adding 2024–2025 scorers would extend the table, but it would not change the paper's thesis. Our contribution is a framework that any future method, including foundation-models-based ones, can and should be measured against. We agree that this is an important direction. In particular, we clarified the scope of our work in the revised introduction and related work.
>
> On foundation models specifically, we would note a scope distinction. Our emphasis is on architecture transferability: ranking many architectures pre-trained on a common source dataset, reflecting how practitioners actually choose among  ImageNet pre-trained models on Hugging Face and PyTorch for fine-tuning. Foundation models introduce new challenges, such as having a per-modality scorer, and revisiting what `"transferability" even means across heterogeneous pre-training sources and objectives, and most existing scorers would need to be redesigned to apply.
>
> That being said, given the rebuttal window, we are not able to run a full foundation-model study with the rigor our own benchmark demands and we would rather not add under-powered experiment that violates the standards we argue for. We added a paragraph positioning foundation-model transferability as the natural next application of our framework.
>
> ---
>
> ### **RC-2: The weighted tau metric is the only one used for the evaluations.**
>
> We consider the weighted tau as the correlation index because it is the most common choice in transferability estimation, based on our literature review (Table 4, Appendix K - Bench. Column). Even though it is the most common, there is no consensus in the literature on which correlation to use, and the choice can widely affect the results, since each correlation index reflects different prior assumptions about how to construct the ranking. In addition, we discuss the proposed combined tau and highlight its distribution across all experiments in our design (Figure 2).
>
>
> ---
>
> ### **RC-3: Sections 5.3 and A8 compare the various scorers, but I find the wording regarding “data” and “dataset” confusing, since the same datasets are being used.**
> We clarified the distinction between the two terms and changed the wording from "data" to "calibration tuples," which more accurately conveys the meaning in this context. In that case, "data" refers to the different data-tier quality used to train the combination strategies (Section 4), while we use calibration tuples as described in Section 4.1. Additionally, we improved the notation in Section 4.1 and enhanced the description of the training evaluation protocol in Section 4.2.
>
> ---
>
> ### **RC-4: Ablation on the number of  experiments/datasets necessary to achieve a good transfer score.**
> Section 5.3 discusses how calibration tuples affect the performance of the proposed combined scorer. Note that our method is not meant to be used as a standalone solution since it requires those measures as inputs. Ours imposes no constraints on the number of available scorers or datasets. Figure 10 (Appendix H) shows performance for different numbers of calibration scorers per combination strategy. The general trend indicates that average performance increases with the number of scorers, except for the linear pooling methods. Results indicate that 4+ scorers are sufficient to achieve good transfer performance across many strategy combinations, and that more scorers help stabilize the mean performance. Note that we evaluate all pair-wise combinations in this plot, so there will be some combinations that are naturally better/worse than others.
>
> Table 3 (Appendix I) shows the runtime for each combination strategy (Section 5.3). We show that 1) the number of scorers' measurements has minimal impact on the training time, scaling from 3 scorers (bot/top performers to 7 (mid performers); and 2) the training time for the worst case is under 120 seconds on a CPU. In all cases, all combinations are cheap and efficient to train.
>
> ---
>
> ### **RC-5: Add references to all sections of the appendix.**
> We enhance the reference across the appendix, including when we list the datasets we use.

---

### Review · Reviewer_XPtR · 2026-05-26

**Summary Of Contributions:**

In this work, the authors propose a systematic framework for benchmarking transfer learning scorers, which are methodologies designed to estimate a priori the post-fine-tuning performance of a specific architecture on a target dataset. They introduce a simple baseline that utilizes the performance of an architecture on ImageNet as a predictive score. Despite its simplicity, this baseline outperforms the vast majority of existing scoring methods in the literature, underscoring a significant limitation in current research. Furthermore, the authors propose the adoption of combination strategies to develop an ensemble scorer capable of exploiting the complementary strengths of various scoring functions. The study evaluates these methodologies across 10 architectures, 13 transferability scorers, and 11 datasets.

**Audience:**

Yes

**Audience Explanation:**

The investigated domain is peculiar and yet relevant. As captured by the opening sentence of the abstract, "In the evolving landscape of deep learning, selecting the best pre-trained models from a growing number of choices is a challenge," the authors highlight a fundamental reality of both contemporary and future deep learning paradigms.

**Broader Impact Concerns:**

There are no broader impact concerns.

**Claims And Evidence:**

Yes

**Claims Explanation:**

Although certain revisions are necessary to improve the presentation of the manuscript and to refine or expand upon the experimental evaluation, the core claims of the submission are convincing and supported by evidence.

**Requested Changes:**

Sections 3, 4, and 5 need to be expanded or polished in certain parts to make the message clearer. Specifically:

* $A_{opt}^i$ should have $i$ as a subscript to be consistent with previous use, and $P(T, A_i)$ should be written as $P(T, A_i^{opt})$.
* The paragraph "Quality evaluation of traditional scorer" has some issues. Specifically, what do the indices $i$ and $j$ stand for? Does $x$ contain the scores of $n$ different scorers? And shouldn't $y$ have the same result of fine-tuning performance to compare the $n$ scores with? Additionally, the function $\rho_z(z_i)$ is unclear; it is not present in Equation 2.
* The subsequent paragraph regarding the combined analysis continues to have some clarity problems in its explanation. Are the groups of data $N$ the ensemble of datasets? Is the "complete set of samples" a dataset? What is the cardinality of each such set? A more in-depth explanation of the dimensionality is important here.
* The phrase, "We apply Z-normalization (subtracting the group mean and dividing by the group standard deviation) per (d, s) tuple groups to ensure comparability across heterogeneous scorers and datasets by subtracting the mean and dividing by the standard deviation," contains repetition and can be stated more concisely.
* In the overall strategy, $t_{test}$ is not properly introduced, and it remains unclear throughout the rest of the explanation. The explanation can also be expanded a bit to improve clarity. Do the authors "pool" the calibration data in the sense that they construct a data pool made of transferability scores as data points and real fine-tuning performances as associated labels?
* Section 4.4 is needed earlier in the text to ensure a better understanding, or the information contained within it should be integrated into the previous text.
* The bootstrap resampling needs a better explanation of how it was used. Additionally, while it is true that a new model architecture requires hyperparameter fine-tuning, this should presumably be done to produce the optimal accuracy metric to compare the scorer score against. I miss the authors' point here.

Regarding expanding the evaluation methodologies:

* The authors should try combination strategies that go beyond a single logistic regression in the univariate strategies; even a simple MLP can contribute to enlarging such an assessment.
* In the univariate strategy, why did the authors not include one accounting for different architectures too, for example, coupled with the "per scorer" strategy, leaving the datasets out, or in other combinations?
* In the SVR, what kind of kernel is employed? Did the authors test various kernels?
* I would also like to see an enlargement of the transformer architectures tested, like the ViT-B, for example. This is suggested given that ResNet-50 is present, as I would advise the ViT-T due to mobile architectures being involved. I acknowledge that the authors have already included a large combination of datasets, architectures, and transferability scores. Thus, I consider this a minor point.
* Importantly, I would like to see results without PACTran among the top scorers, given that it yields an average of 38 in Figure 2. PACTran should not be considered a top scorer given those results.

Some minor points:

* Some labels in Figure 1 are too small and should be enlarged.
* The phrase "First, we recognize the absence of a strong baseline scorer" is too bold and needs to be better integrated within the context.
* The notation in Section 4.1 would be more useful if better integrated into the text when each of these symbols is needed, especially given that many of them are used just once.
* Regarding "Garbage in, gold out," the word "garbage" is too informal and should be replaced.
* I suggest the authors employ subsequent lettering for the structure of the supplementary material (e.g., Appendix A, B, C, etc.) rather than nested formatting like A.1, A.2, and so on.

---

> ### Author Response · Authors · 2026-06-13
>
> We appreciate the reviewer's thoughtful and constructive suggestions. Below, we respond to the major requested changes.
>
> ---
>
> ### **RC-{1,2,4,5,6}: Notation and clarity in Sections 3–5**
> We revised Sections 3–5 for clarity. We moved the training/evaluation protocol to Section 4.2 and standardized notation (e.g., $ P(T, A^{opt}_{i}) ) $.
>
> We now state that indices $i,j$ in Equation 2 denote model architectures; $\mathbf{x}$ holds a single scorer's values across architectures and $\mathbf{y}$ the matching fine-tuned performances; and $\rho_{\mathbf{z}}(z_i)$ denotes the rank of $z_i$ in vector $\mathbf{z}$ (Section 3).
>
> We added an explicit paragraph that explains the grouped (per-dataset) structure, the cardinality of each group, the pooled statistic, and a concise Z-normalization sentence. Changes appears in Section 3, Section 4.2, and the start of Section 5.
>
> ---
>
> ### **RC-7: Bootstrap resampling and fine-tuning protocol**
> We clarified bootstrapping in Section 5. For each dataset and scorer, we bootstrap paired (score, performance) observations across the $A$ architectures, recompute the correlation on each resample, and report point estimates plus uncertainty intervals. We use 1000 resamples, compute the statistic on each, and report the mean and 95% percentile intervals (distributions in figures 2 and 4) to make robust estimates (Section 5.1).
>
> ---
>
> ### **RC-{8-10}: Combination strategies, SVR kernel, and non-linear models**
> Section 4 now outlines the combination strategies we evaluate, including Linear per-Scorer, Linear per-Scorer/Dataset, Least Squares, and SVM. For our SVM implementation, we use a Gaussian kernel. We did not explore a wide range of SVR kernels. Initially, we did not consider MLPs because they require tuning several additional parameters, such as the number of epochs, learning rate, number of layers, optimizer, and activation functions. Most of the combination strategies here do not require parameter search. Our findings indicate that simple linear models can outperform many existing methods in the literature, depending on the employed combination strategy.
>
> ---
>
> ### **RC-12: Moving PACTran from Top to Mid scorers**
> We have included an ablation study regarding "Moving PACTran" in the supplementary material (Appendix L). Our findings demonstrate that relocating PACTran to the mid-tier does not alter the paper's primary conclusions. In fact, the updated results reveal improved performance gains across both the top and mid data tiers (as illustrated in Figure 13) and across all combination strategies.
>
> ---
>
> ### **RC-{13-17}: Presentation and minor points**
> We addressed all minor points suggested in the new version of the text. We replaced informal wording (Section 5), better integrated the baseline claim into the introduction (Section 1), clarified notation in Section 4.1, and relettered the appendix (Appendix A, B, ...).

---

> > ### Comment · Reviewer_XPtR · 2026-06-17
> > **Final Comment**
> >
> > As the authors have addressed the majority of my concerns, I recommend the manuscript for acceptance. However, I note that in the final version, the authors should explicitly exclude PACTran from the list of top-performing scorers, rather than simply relegating the related results to the supplementary materials. Given that the results presented in Figure 2 clearly demonstrate inferior performance of PACTran compared to several other evaluated scorers, including it among the top performers is counterintuitive.

---

### Review · Reviewer_AYbR · 2026-05-30

**Summary Of Contributions:**

This paper studies the benchmarking transferability scorers for selecting pre-trained models before expensive fine-tuning. The paper proposes a benchmark over 10 vision model architectures, 13 transferability scorers, and 11 image classification datasets, and introduces ImageNet top-1 accuracy as a simple but strong baseline. The paper further proposes an aggregated weighted Kendall’s tau for combined analysis across datasets, and a combined scorer that learns to aggregate multiple individual scorers. Overall, the paper is clearly motivated and addresses an important practical issue in model selection for transfer learning.

**Audience:**

Yes

**Audience Explanation:**

I think generally the audience working on general machine learning (o.o.d., transfer learning, few-shot learning) or the vision domain would be interested in this work.

Specifically, the proposed aggregated analysis is also useful for reporting scorer performance dataset by dataset can become difficult to interpret, and a unified statistic that preserves within-dataset ranking comparisons is a reasonable contribution. The rank-rank visualizations are intuitive and help communicate when a scorer consistently matches the true transfer performance ordering.

**Claims And Evidence:**

Yes

**Claims Explanation:**

1. The paper does a good job benchmarking the question. The authors explicitly revisit ImageNet accuracy as a baseline. The finding that this simple baseline outperforms many existing transferability scorers is valuable, and it provides a useful sanity check for future work in this area.

2. The paper covers a reasonably diverse set of model families and target datasets, including generalist, fine-grained natural, artifact, and medical datasets. This makes the empirical message more convincing than a small-scale comparison on only a few standard vision datasets.

3. The combined scorer is also an interesting and practically useful direction. Instead of treating existing scorers as mutually exclusive alternatives, the paper asks whether their signals can be combined. The result that simple calibration or linear combination can outperform individual scorers is useful, especially because it suggests that future transferability estimation may benefit from combining complementary signals rather than designing a single universal metric. The ablation on scorer quality shows that the combined scorer is not simply “garbage in, gold out” -- the quality of the base scorers matters.

**Requested Changes:**

1. Clarify the scope of “transfer learning.”: The paper is motivated as a broad benchmark for transfer learning, but the actual setting is mainly architecture-level transferability estimation for ImageNet-pretrained image classification models with supervised fine-tuning. Since transfer learning now includes many other regimes, such as test-time adaptation, in-context/prompt-based transfer, few-shot adaptation, parameter-efficient fine-tuning, and zero-shot/foundation-model transfer, the authors should either narrow the framing, or explicitly discuss how their benchmark relates to these broader settings.

2. Based on the point above, it would be nice to add evaluations under additional transfer protocols, if possible. To better support the broader claims, it would be important to include at least one additional transfer protocol beyond full supervised fine-tuning.

3. The current benchmark is useful, but it remains limited to image classification. I encourage the authors to add a small experiment on another domain or modality, such as text classification with pretrained language models, vision-language zero-shot/few-shot transfer, object detection, segmentation, or multimodal retrieval. Or at least discuss these.

---

> ### Author Response · Authors · 2026-06-13
>
> We thank the reviewer for your comments and feedback. We appreciate you found the topic clearly motivated and that you think it addresses an important practical issue. We answer your questions and concerns below.
>
> ---
>
> ### **RC-1: Scope clarification**
> Our work aims to provide a broad benchmark of transfer learning through the lens of the transferability estimation literature for image classification tasks. This literature focuses on developing transferability scores that predict how well a model will perform when applied to a specific target dataset (Section 2). Our scope focuses on architecture transferability, given the large number of open-source model architectures pre-trained on the same source dataset available on large model hubs. That said, our contributions include: 1) systematic, large-scale benchmark for evaluating transferability scorers and validate it across 10 vision model architectures, 13 transferability scorers, and 11 datasets; 2) show that the performance of pre-trained models on ImageNet is a simple and robust baseline, surprisingly outperforming most of the evaluated scorers; 3) show that the potential of using linear models to aggregate information from multiple base scorers.
>
> ---
>
> ### **RC-2: Additional transfer scenarios**
> The scorers we study in Appendix K largely depend on full fine-tuning signals and, in several cases, on class-label information (LEEP, NCE, GBC, to name a few), so they do not transfer cleanly to other adaptation protocols, such as image segmentation, semantic segmentation,. We therefore keep the protocol fixed in Section 3 to isolate scorer behavior rather than confound it with a change in adaptation strategy. Note that while our proposed methodology in Section 3 can accommodate tasks beyond classification, the scorers are limited to scenarios within this specific context.
>
> ---
>
> ### **RC-3: Benchmark limitation to image classification tasks**
>
> Given these limitations, adapting these methods for tasks beyond classification often requires redesigning each approach, incurring engineering effort and, in some cases, a complete redesign of each scorer. As many prior works struggled with image classification (their main task), we found it premature to explore scenarios beyond classification and full-fine-tuning.  Nonetheless, our benchmark study considers dataset categories, including natural, fine-grained, artifact, and medical images (Section 5), so we test the scorer's capabilities under several shifts. This design lets us study transferability in a controlled setting rather than mixing it with a second transfer problem. Given the rebuttal window, we are not able to run a full study that considers tasks beyond classification with the rigor that our benchmark demands.

---

### Author Response · Authors · 2026-06-13
**Revised Manuscript**

We thank all reviewers for carefully reading our manuscript and providing constructive feedback. We posted detailed responses to all reviewers and a revised version of the manuscript that incorporates new experiments and many reviewers' feedback. All changes are marked in **red** for easy identification.

---

### Decision · Action_Editor_KeoL · 2026-07-10

**Recommendation:** Accept as is

**Additional Comments:**

One of the reviewers mentions the request to exclude PACTran from the list of top-performing scorers as the results in Figure 2 demonstrate its inferior performance.

**Audience:**

Yes

**Audience Explanation:**

In the absence of state-of-the-art evaluation protocols, this paper provises a well-needed benchmark, which could be of interest to the community.

**Claims And Evidence:**

Yes

**Claims Explanation:**

The authors propose a systematic benchmark for transferability scorers. The large-scale empirical study evaluates thirteen scorers across ten vision model architectures and eleven datasets. In addition to the evaluation, they show that a novel combined scorer  can outperform multiple individual ones.

All the reviewers point out that the authors have addressed comments, including rewriting some parts of the paper to improve readability. One of the reviewers notes that some additional experiments could further strengthen the paper. Overall, it seems to be a good benchmark work.